# Convergence-Aware Online Model Selection with Time-Increasing Bandits

## ABSTRACT

Web-based applications such as chatbots, search engines and news recommendations continue to grow in scale and complexity with the recent surge in the adoption of large language models (LLMs). Online model selection has thus garnered increasing attention due to the need to choose the best model among a diverse set while balancing task reward and exploration cost. Organizations faces decisions like whether to employ a costly API-based LLM or a locally finetuned small LLM, weighing cost against performance. Traditional selection methods often evaluate every candidate model before choosing one, which are becoming impractical given the rising costs of training and finetuning LLMs. Moreover, it is undesirable to allocate excessive resources towards exploring poor-performing models. While some recent works leverage online bandit algorithm to manage such exploration-exploitation trade-off in model selection, they tend to overlook the increasing-then-converging trend in model performances as the model is iteratively finetuned, leading to less accurate predictions and suboptimal model selections.

In this paper, we propose a time-increasing bandit algorithm TI-UCB, which effectively predicts the increase of model performances due to training or finetuning and efficiently balances exploration and exploitation in model selection. To further capture the converging points of models, we develop a change detection mechanism by comparing consecutive increase predictions. We theoretically prove that our algorithm achieves a lower regret upper bound, improving from prior works' polynomial regret to logarithmic in a similar setting. The advantage of our method is also empirically validated through extensive experiments on classification model selection and online selection of LLMs. Our results highlight the importance of utilizing increasing-then-converging pattern for more efficient and economic model selection in the deployment of LLMs.

## CCS CONCEPTS

• **Computing methodologies → Search with partial observations**; **Online learning settings**.

## KEYWORDS

Model Selection; Online Learning; Multi-Armed Bandit; Large Language Model

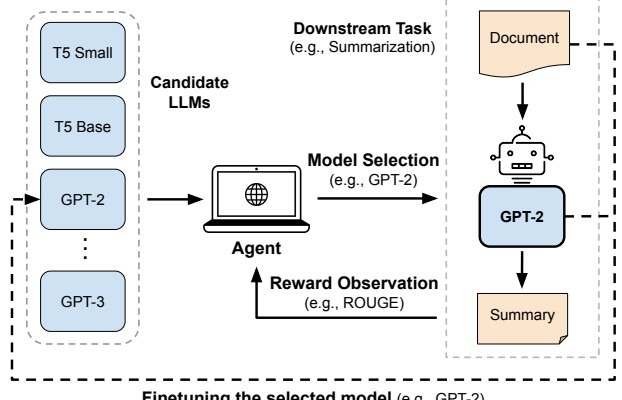

**Figure 1: An illustrative example of online model selection for LLM summarization.**

**ACM Reference Format:**

Anonymous Author(s). 2018. Convergence-Aware Online Model Selection with Time-Increasing Bandits. In *Proceedings of Make sure to enter the correct conference title from your rights confirmation emai (Conference acronym 'XX).* ACM, New York, NY, USA, 15 pages. https://doi.org/XXXXXXX.XXXXXXX

## 1 INTRODUCTION

Recent times have witnessed the promising adoption of large-scale pretrained models such as large language models (LLMs) [6, 9, 35, 39, 40] in various web-based applications including online chatbots, search engines and news recommendations. While different models may exhibit different tasks-specific advantages, the increasing number of high-performing models particularly LLMs recently has resulted in growing interest in the online model selection problem [15, 16, 23, 28, 33]. The goal of online model selection is to choose the best model from a diverse set, maximizing the reward on a specific task while minimizing the exploration cost in the selection process, e.g., the economic trade-offs of LLMs [21].

For example, organizations today face decisions like whether to employ a costly API-based LLM or a local small LLM being fine-tuned over time. The choice hinges on factors such as cost-effectiveness and robust performance in practical applications [21]. API-based LLMs [6, 35], while offering impressive zero-shot performance, typically charge based on usage. In contrast, small locally-maintainable LLMs [39, 40] would be much cheaper for heavy usage and potentially more performant after being sufficiently finetuned.

Existing LLM selection schemes [37, 38, 42] primarily exploit the best-performing model in a static setting, e.g. choosing the LLM that generates answers with lowest perplexity score at the current state [38]. Prior works on more general model selections lie in parameter-free online learning [11, 15, 36] and in the field of Automated Machine Learning (AutoML) [13, 20, 24, 28, 31], e.g. treating

the selection of model as a new hyperparameter to be optimized by Bayesian optimization [13, 24]. These schemes, though effective, often require full information or comprehensive evaluation of models. This may be impractical for recent large-scale models such as LLMs, where training or finetuning could be costly and thus spending excessive resources on exploring poor-performing models is undesirable. Thus a trade-off between exploration, i.e. learning about models' performances being trained or finetuned, and exploitation, i.e. selecting the best-performing one currently, is needed. Some recent works [8, 23, 28, 33] have proposed to manage the trade-off in online model selection by formulating it as a multi-armed bandit problem, which is similar to our problem setting. However, they tend to overlook the increasing-then-converging trend in model performances as models are iteratively trained or finetuned when they are selected, which is illustrated by the process shown in Figure 1 and the curve in Figure 2. As a result, these methods often make inaccurate predictions on future model performances and thus make suboptimal selections of models.

To address the above limitations, we propose an increasing bandit algorithm, Time-Increasing UCB (TI-UCB), which can promisingly predict and capture the increasing-then-converging pattern of model performances and efficiently balances the exploration and exploitation in online model selection. Specifically, TI-UCB adopts a least square estimator to piecewise-linearly approximate the increasing trend from past reward observations. To further capture the converging points, we develop a sliding-window change detection mechanism by comparing consecutive increase predictions. We provide a theoretical analysis of our proposed method and prove TI-UCB achieves a lower regret upper bound, improving prior works' polynomial regret [33] to logarithmic regret in an increasing-then-converging bandit problem. We also empirically validate the advantage of our method in terms of performance and parameter robustness through synthetic experiments and real-world experiments of online model selection for canonical classification models as well as recent LLMs.

In summary, we make the following major contributions:

- Motivated by the need for efficient and economic online model selection (e.g., selecting the best-performing LLMs being finetuned while minimizing the cost), we formulate it as time-increasing bandit problem with increasing-then-converging trend to balance the exploration and exploitation.
- Capitalizing the increasing-then-converging model performance trend, we propose the TI-UCB bandit algorithm, which can promisingly predict the reward increase and capture the converging points with a change detection mechanism comparing consecutive increase predictions.
- We theoretically prove that TI-UCB achieves a lower regret upper bound, improving prior works' polynomial regret to logarithmic in a similar increasing bandit problem.
- We empirically validate the advantages of TI-UCB through extensive experiments of online model selection for classification models as well as recent LLMs. Our results highlight the importance of utilizing increasing-then-converging pattern for more efficient and economic model selection in the deployment of LLMs.

## 2 RELATED WORK

In this work, we mainly focus on the online model selection formulated as a multi-armed bandit problem, which is closely related to the following two lines of works.

### 2.1 Online Model Selection

Online model selection has been drawing attention in choosing the best one among the increasing number of high-performing models, e.g. LLMs, with limited training resources and performance evaluations [15, 16, 23, 28, 33]. Existing LLM selection schemes [37, 38, 42] primarily focus on static model selection without considering the model performance change due to iterative finetuning. For example, Peng et al. [38] choose the LLM that generates texts with lowest perplexity score at the current state [38]. While several prior works on more general model selections have studied the parameter-free online learning [11, 15, 36], most of them assume full information in the online setting, which could be impractical when training and evaluation costs are high and thus only the feedback from the selected model might be observable. Some recent works have explored online model selection with partial information. Foster et al. [16] consider the model selection problem with contextual bandit feedback. Cella et al. [8] formulate online model selection as a rested bandit problem. Karimi et al. [23] utilize active learning to select the best model among a pool of pre-trained classifiers.

The online model selection problem is also closely related to the wider field of AutoML, whose objective is to automate the entire process of applying machine learning to real-world problems [14, 22]. A similar application of AutoML to online model selection is hyperparameter optimization [13, 20, 24, 28, 31]. The commonly adopted Bayesian optimization frameworks [13, 24] for hyperparameter selections treat the selection of the model as a new hyperparameter to be optimized. While these approaches are faced with problems of inefficiency resulting from a huge parameter space, recent works [12, 27, 28, 33] have been leveraging bandit algorithms to balance exploration and exploitation for more efficient online model selection. While many approaches have been proposed as listed above, most of them consider a static setting and overlook the increasing-then-converging model performance as models are trained or fine-tuned alongside the model selection process. In comparison, our work emphasize and utilize the such increasing-then-converging pattern due to model finetuning when selected, as illustrated in Figure 1.

### 2.2 Non-stationary Bandits

The increasing-then-converging reward trend poses the challenge of non-stationarity in online model selection, which is also closely relaetd to non-stationary bandit problem. Non-stationary bandits typically feature rewards that are either piecewise-stationary [2, 4, 7, 18, 30, 46] or smoothly-changing [3, 5, 41]. In piecewise-stationary environments, the reward distribution is piecewise-constant and changes abruptly at unknown time points. Existing methods often use certain selection criteria for recent observations [2, 18, 46] or change-detection techniques to focus on observations after change points [4, 7, 30]. Garivier and Moulines [18] propose D-UCB and SW-UCB, with the former using a discount factor for past rewards and the latter employing an adaptive sliding window for recent

observations. Cao et al. [7] apply a sliding window to detect abrupt changes, while Liu et al. [30] use cumulative sums for change point detection and UCB index updates. Smoothly-changing environments, on the other hand, describe situations where rewards change continuously. Besbes et al. [3] propose Rexp3, a modification of the Exp3 algorithm for adversarial MABs, based on known reward variation. Bouneffouf and Féraud [5] and Russac et al. [41] develop UCB-based algorithms for known reward changing trends. Trovo et al. [46] present SW-TS, a Thompson Sampling-based algorithm with a sliding window, for environments with smoothly-changing rewards. Though considering non-stationary, these works have no guaranteed performance in cases of increasing-then-converging reward trends exhibited in online model selection problem.

Our setting focuses on rested bandits, where arm rewards depend on the number of times the arm is pulled, e.g., the model performances depend on the number of times the model is trained or finetuned. This setting, introduced by [45], has been further explored in a subcategory called rested rotting bandits [26, 43, 44], where rewards decrease with each pull. Recently, there has been discussion on the counterpart setting with increasing rewards [8, 19, 28, 33]. However, existing approaches still have certain limitations. For example, Heidari et al. [19] assume accurate reward feedback without stochasticity, while Li et al. [28] solve the hyperparameter optimization problem in a similar deterministic setting. Although Cella et al. [8] consider a non-deterministic increasing reward scenario, they assume knowledge of the parametric form of the reward. Among the approaches above, Metelli et al. [33] consider stochastic bandits with non-decreasing and concave rewards and utilizes a sliding window to focus on most recent observations, which is the mostly closely related work to our setting. However, such recent-observation-based method may not be able to accurately capture the increasing-then-converging pattern exhibited in online mode selection problem, and thus lead to suboptimal reward predictions and slow reactions to converging points. In comparison, our proposed TI-UCB predicts the increasing reward and adaptively capture the converging point, which demonstrates advantages over R-ed-UCB both theoretically and empirically in online model selection problem.

## 3 PROBLEM FORMULATION

In this section, we first introduce the setting in online model selection, which is similar to [8, 23, 28, 33]. Then, we formulate it as time-increasing bandits with increasing-then-converging trend, where we novelly capture the model performance trend.

### 3.1 Online Model Selection

The online model selection process can be described with the following three steps forming a feedback loop. For consistent illustration as in Figure 1, we use the online selection of LLMs for text summarization as describing examples for each step below.

*Model Selection.* Among a set of candidate LLMs, the agent first selects an LLM to deployed for a document summarization task according to a selection policy.

*Reward Observation.* Given a test document, the selected LLM generates a summary of the document. Comparing the generated

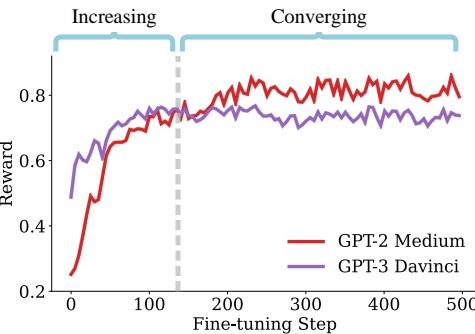

**Figure 2: Increasing-then-converging reward trends of an API-based LLM (GPT-3 Davinci) and a local small LLM (GPT-2 Medium) over finetuning steps on a text summarization dataset. The reward considers both model performance and API finetuning cost, details of which can be found in Section 5.4. GPT-2 Medium is observed to outperform GPT-3 Davinci after certain finetuning steps and hence such reward trends make it non-trivial to apply existing methods for online model selection.**

summary with the reference summary, an evaluation score measuring summarization quality, i.e., model performance, is computed. The agent then observe the evaluation score as the reward of the previous model selection and correspondingly update its selection policy for next round of model selection.

*Model Finetuning.* After the agent receives the reward, the selected LLM is fine-tuned with the test document-summary pair in previous step. The finetuned LLM will then be in the candidate pool for the next round of model selection as in the first step.

The goal of the agent is to efficiently select the best LLM being finetuned, yielding the highest cumulative reward in a long run.

### 3.2 Time-Increasing Bandits with Increasing-then-Converging Trend

We formulate the online model selection as a time-increasing bandit problem with increasing-then-converging trend, capturing the model performance trend overlooked in previous bandit formulations [8, 23, 28, 33]. Suppose there are $K$ arms (e.g., candidate LLMs) in the environment. Without loss of generality, we denote $i$ to be the $i$-th one among $K$ arms. The agent will interact with the environment for $T$ rounds. At each time step $t$, the agent pulls an arm $A_t \in [K]$ and observes a reward $X_{A_t,t}$, which is a random variable drawn from a probability distribution with expectation $\mu_{A_t}(t)$. Note that in our setting, the expected reward $\mu_i(t)$ of arm $i$ at time $t$ depends on the number of times that $i$ is pulled (e.g., the LLM performance depends on the number of times the model is fine-tuned). Thus, denoting $n_i(t) = \sum_{s=1}^{t} \mathbf{1}\{A_s = i\}$ as the total number of pulls on arm $i$ till the end of time step $t$, we define the reward of arm $i$ at time step $t$ as

$$\mu_i(t) = \mu_{i,n_i(t)} \ .$$

Similarly, we define the empirical mean on arm $i$'s expected reward at time step $t$ as

$$\hat{\mu}_i(t) = \hat{\mu}_{i,n_i(t)} = \frac{1}{n_i(t)} \sum_{s=1}^{t} \mathbb{1}\{A_s = i\} X_{A_s,s} \, . \tag{1}$$

Suppose the expected reward of arm $i$ first increases with the number of its pulls until it reaches the stable stage (e.g. The LLM performance increases with the number of times it is finetuned until converging). Denote $v_i$ as the converging point that the reward of arm $i$ becomes stable. The value of $v_i$ could be different across arms (e.g., LLMs may have different convergence rate when finetuned). Note that $v_i$ is unknown to the agent and is a value that needs to be learned by the agent. A typical challenging time-increasing bandit example is shown in Figure 2, where GPT-3 Davinci gives higher reward with strong zero-shot performance but is outperformed by GPT-2 Medium due to its high finetuning cost despite the performance improvement.

The goal of the agent in time-increasing bandit is to maximize the cumulative reward in a long run, which is equivalent to minimize the cumulative regret. We define the cumulative expected regret as

$$\mathbb{E}\left[R(T)\right] = \sum_{i=1}^{K} \sum_{s=1}^{n_i^*(T)} \mu_{i,s} - \sum_{i=1}^{K} \mathbb{E}\left[\sum_{s=1}^{n_i(T)} \mu_{i,s}\right], \tag{2}$$

where $n_i^*(T)$ is the total number of times arm $i$ is pulled in the optimal action sequence that maximizes the reward with a greedy strategy and $n_i(T)$ is the actual number of times that arm $i$ has been selected by the agent till time step $T$.

## 4 PROPOSED METHOD

In this section, we first introduce the proposed Time-increasing UCB (TI-UCB) algorithm in Section 4.1 to solve the online model selection problem formulated in Section 3. Then in Section 4.2 we theoretically analyze the regret upper bound of TI-UCB in a typical increasing-then-converging setting.

### 4.1 TI-UCB Algorithm

The TI-UCB is described in Algorithm 1. Note that in our settings, the reward of each arm is non-stationary and changes each time the arm is pulled. Therefore, to maximize the cumulative reward and thus the cumulative regret, the optimal arm should be chosen and explored at the early stage, followed by further exploitation at the later stage. Since the reward will later reach a stable state at $v_i < T, i = 1, \cdots, K$ as defined in Section 3.2 and observed in Figure 2, e.g., the increasing-then-converging performance trend of LLMs, the algorithm also aims to detect the change points of each arm.

To achieve the above goals, TI-UCB comprises two primary phases. The first phase is the increase prediction process with an upper confidence bound, which corresponds to Line 2-8 of Algorithm 1. The second phase is the change detection process, which corresponds to Line 9-13 of Algorithm 1. We describe the two phases in details as below.

#### 4.1.1 Increase Prediction.
Recall that $n_i(t)$ is the number of pulls on arm $i$ by the end of time $t$. We approximate the initial increasing

---

**Algorithm 1** TI-UCB

**Input:**
    $K, \delta$, window size $\omega$, threshold $\gamma$;
**Output:**
    **Initialize:** $\tau_i' \leftarrow 1, n_i \leftarrow 0, \forall i \in [K]$;
1: **for** $t = 1, ..., T$ **do**
2:     **for** $i = 1, ..., K$ **do**
3:         $\bar{\mu}_{i,n_i(t)} = \hat{\mu}_{i,n_i(t)} + 16\sqrt{\frac{2\ln(1/\delta)}{n_i(t)}}$;
4:     **end for**
5:     Pull arm $A_t \leftarrow \arg\max_i \bar{\mu}_{i,n_i(t)}$;
6:     Observe reward $X_{A_t,t}$;
7:     Update estimation $\hat{\mu}_{i,n_i(t)}$;
8:     Update number of pulls $n_{A_t}(t) \leftarrow n_{A_t}(t) + 1$;
9:     **if** $n_{A_t}(t) \geq 2\omega$ **then**
10:         **if** $|\hat{\mu}_{w_1,A_t}(t+1) - \hat{\mu}_{w_2,A_t}(t+1)| > \frac{\gamma}{2}$ for arm $A_t$ **then**
11:             $\tau_{A_t}' \leftarrow t$ and $n_{A_t}(t) \leftarrow 1$;
12:         **end if**
13:     **end if**
14: **end for**

---

trend of each arm's reward as

$$\hat{\mu}_{i,n_i(t)} = \hat{a}_{i,n_i(t)} \cdot n_i(t) + \hat{b}_{i,n_i(t)} \, ,$$

where $\hat{a}_{i,n_i(t)}$ is the approximated reward growth rate of arm $i$ and $\hat{b}_{i,n_i(t)}$ is the intercept term, both of which are calculated using the least square method and linear regression with observation records. In each time step $t$, the algorithm first updates $\hat{a}_{i,n_i(t-1)}$ and $\hat{b}_{i,n_i(t-1)}$ based on previous reward observations, and then predicts the reward of current time step $\hat{\mu}_{i,n_i(t)}$ based on the estimated increasing trend.

With increased reward prediction, TI-UCB seeks to balance exploration and exploitation by adding an uncertainty term to the predicted reward of each arm as Line 3 of Algorithm 1. The concentration level of the approximated coefficients in the uncertainty term is derived from Proposition 1. The algorithm then chooses the arm $A_t$ with the maximum upper confidence value $\bar{\mu}_{i,n_i(t)}$, receives a reward $X_{A_t,t}$ and update the observation records of arm $i$ as decribed in Line 5-7 of Algorithm 1. Note that to simplify the notation, we use $\hat{\mu}_{i,n_i(t)}$ as $\hat{\mu}_i(t)$, and $\bar{\mu}_{i,n_i(t)}$ as $\bar{\mu}_i(t)$ in the rest of the paper.

**Proposition 1.** *The upper confidence bound in TI-UCB for arm $i$ can be defined as*

$$\bar{\mu}_i(t-1, \delta) = \begin{cases} \infty, & \text{if } n_i(t-1) = 0 \, , \\ \hat{\mu}_i(t-1) + 16\sqrt{\frac{2\ln(1/\delta)}{n_i(t-1)}}, & \text{otherwise,} \end{cases}$$

*Then for any $\delta \in (0, 1), \mu \leq \hat{\mu} + 16\sqrt{\frac{2\ln(1/\delta)}{n}}$ holds with probability at least $1 - \delta$. Detailed proof is provided in Appendix A.1.*

#### 4.1.2 Change Detection.
After certain amounts of arm pulls, different arms will reach a stable state with stable reward $c_i, i \in [K]$

at different time steps. To infer change points $v_i$ based on the observed rewards of each arm $i$, we set two sliding windows $w_1$ and $w_2$ each with length $\omega$ to monitor rewards along the timeline, moving forward with more reward are observed for arm $i$.

To detect reward changes, we compare the predicted reward at time step $t + 1$ calculated based on the previous window of reward observations from $w_1 = [n_i(t) - 2\omega + 1, n_i(t) - \omega]$, and based on the current window $w_2 = [n_i(t) - \omega + 1, n_i(t)]$, which we refer to as $\hat{\mu}_{w_1,i}(t+1)$ and $\hat{\mu}_{w_2,i}(t+1)$ respectively. If the difference of the two predictions exceeds the preset threshold $\gamma/2$ as described in Line 10 in Algorithm 1, we consider a reward change point of arm $i$ is detected and the observation records of arm $i$ will be re-initialized and the current time step will be recorded as $\tau_i'$ representing a change point. Otherwise, the algorithm continues to pull next arms, detecting change points with new reward observed. The rational of change detection is formalized in Proposition 2.

**Proposition 2.** *The reward change point of arm $i$ is considered to be reached, if*

$$|\hat{\mu}_{w_1,i}(t+1) - \hat{\mu}_{w_2,i}(t+1)| > \gamma/2 \, ,$$

*where $\hat{\mu}_{w_1,i}(t+1)$ and $\hat{\mu}_{w_2,i}(t+1)$ are the predicted rewards for arm $i$ at time $t+1$ calculated by observations in the window $w_1 = [n_i(t) - 2\omega + 1, n_i(t) - \omega]$, and by observations in the window $w_2 = [n_i(t) - \omega + 1, n_i(t)]$. With $\gamma \leq \sqrt{\frac{2}{\omega}(14 + \frac{12}{|\omega-1|})^2 \ln(\frac{2}{\delta})}$, the above change detection inequality is valid with probability $1 - \delta$. Detailed proof is provided in Appendix A.2.*

### 4.2 Regret Upper Bound of TI-UCB

In this section, we provide the regret upper bound of TI-UCB in a typical increasing-then-converging reward setting as follows and detailed proof can found in Appendix B.

Specifically, we assume the reward received by each arm $i$ first increases linearly with the number of times it is pulled, and then abruptly changes to a stable value $c_i$ within the range of $[0, T]$, where $T$ remains unknown. Such reward trend approximately captures the increasing-then-converging performance pattern of LLMs in online model selection scenarios.

**Theorem 1.** *Assume that $\delta \leq 1/T$, then the expected regret of TI-UCB algorithm is bounded by*

$$\mathbb{E}\left[R(T)\right] \leq \sum_{i:n_i(T) \geq n_i^*(T)} c_i \frac{4096 \ln(T)}{\Delta_{\min}^2} + K\left(\frac{2\pi^2}{3} + \omega + 2 + 2L\right) + 2, \tag{3}$$

*where $\Delta_{\min} = \min_{t \in [0,T], i \neq i_t^*} \{\mu_{i_t^*}(t) - \mu_i(t)\}$ is the minimum gap between the optimal reward and the true reward and $L$ is a constant smaller than $\ln T$.*

*Remark.* The main idea of our proof is to divide the $[0, T]$ period into two stages, $[0, v_i]$ and $[v_i, T]$, where $v_i$ is the reward converging point of arm $i$. Define two events that $F_i = \{\tau_i' > v_i\}$ and $D_i = \{\tau_i' \leq v_i + \omega\}$. $F_i$ implies that the $i$-th change point can only be detected by the algorithm after change really occurs, and $F_i^c$ means that the $i$-th change point is regarded as having occurred but actually change does not happen. Then the regret can be decomposed by

$$\mathbb{E}\left[R(T)\right] = \mathbb{E}\left[R(T)\mathbb{1}\{F_1\}\right] + \mathbb{E}\left[R(T)\mathbb{1}\{F_1^c\}\right] \, . \tag{4}$$

The regret of the first stage is derived using a similar method for proving the regret upper bound for standard UCB algorithm with a generalization to increasing reward. The regret of the second stage involves a further discussion on scenarios of whether the change happened or not with several lemmas introduced, which we refer to Appendix B for more details.

From Theorem 1, we get the regret upper bound of TI-UCB of $O\left(\log(T)/\Delta_{\min}^2\right)$. In comparison, R-ed-UCB [33] achieves under certain condition a regret upper bound of $O(T^{2/3}\log(T)^{1/3})$. Thus, by emphasizing and utilizing the increasing-then-converging reward pattern, TI-UCB improves the prior work's polynomial regret to logarithmic regret in a similar setting.

## 5 EXPERIMENTS

In this section, we evaluate TI-UCB on both synthetic environment and real-world environment of online model selection for canonical classification models and LLMs to validate its empirical performance.

### 5.1 Experimental Setup

In this section, we describe our experiment setup. We first introduce the baselines to compare and the parameter settings. Then we describe the evaluation metrics, followed by the research questions we seek to answer.

*5.1.1 Baselines.* We compare our proposed algorithm against the following baseline algorithms and methods:

- **KL-UCB** [17]: a classic stationary bandit algorithm utilizing KL Divergence.
- **Rexp3** [3]: a non-stationary bandit algorithm based on variation budget.
- **Ser4** [1]: a non-stationary bandit algorithm that takes into account the best arm switches during the process.
- **SW-TS** [46]: a sliding-window bandit algorithm with Thompson Sampling that generally handles non-stationary settings well.
- **SW-UCB** [18]: a sliding-window bandit algorithm with UCB that can handle general non-stationary settings.
- **SW-KL-UCB** [10]: a sliding-window bandit algorithm with KL-UCB.
- **R-ed-UCB** [33]: a recent non-stationary bandit algorithm designed for similar scenarios as ours with non-decreasing and concave rewards.
- **Auto-Sklearn** [13]: the state-of-the-art AutoML system utilizing Bayesian optimization-based solution.

*5.1.2 Parameter Setting and Metric.* We use the recommended parameter settings from the respective papers for all baseline bandit algorithms. Details can be found in Appendix C. For Auto-Sklearn, while the system is designed to automate the model selection given full data in a static setting, it is not directly applicable in online learning setting. Instead, we update the optimizer of Auto-Sklearn every 50 steps with the batched data samples. Note that such approximation to online learning would lead to extra computation cost and time due to frequent optimization steps and thus we only use Auto-Sklearn as a representative AutoML method to be compared.

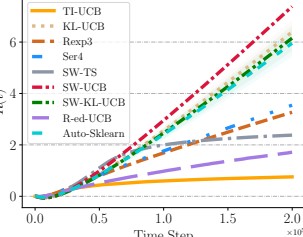
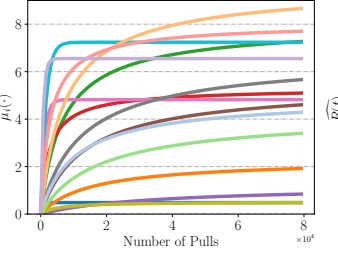
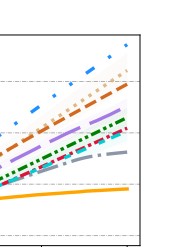
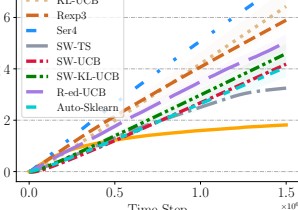

(a) 2-Arm Bandits: Reward Functions  (b) 2-Arm Bandits: Cumulative Regret  (c) 15-Arm Bandits: Reward Functions  (d) 15-Arm Bandits: Cumulative Regret

**Figure 3: Online selection of generated synthetic models covering a variety of increasing-then-converging patterns.**

For our proposed TI-UCB algorithm, we set the change detection window $w = 100$ and threshold $\gamma = 0.3$ in all experiments if not specified otherwise. Note that the setting of parameters may be sub-optimal as we do not optimize all algorithms including TI-UCB.

To evaluate the metric in experiments, we mainly compare the algorithms based on empirical cumulative regret $\widehat{R(t)}$, which is the empirical counterpart of Equation 2 defined as

$$\widehat{R(T)} = \sum_{i=1}^{K} \left[ \sum_{s=1}^{n_i^*(T)} \hat{\mu}_{i,s} - \sum_{s=1}^{n_i(T)} \hat{\mu}_{i,s} \right].$$

All experimental results are averaged over 20 independent runs.

*5.1.3 Research Questions.* The experiments are designed to answer the following research questions:

**RQ1.** Can our proposed TI-UCB algorithm outperform existing methods in online model selection? Is TI-UCB still effective when the increasing-then-converging reward trend are subject to fluctuations?

**RQ2.** Can TI-UCB effectively handle the scenario where we introduce the finetuning cost of API-based LLMs in addition to model performance in reward design, and thus manage the economic tradeoff?

**RQ3.** Is our change detection mechanism really effective in capturing the converging stage? How does different change detection window sizes affect the performance of TI-UCB?

## 5.2 Synthetic Model Selection

In this section, we describe the experimental results on a synthetic environment of online model selection.

*5.2.1 Data Generation.* We generate two sets of synthetic reward functions to simulate models with increasing-then-converging performance patterns. Specifically, we sample respectively 2 and 15 reward functions $\mu_i(\cdot)$ from the following two function families:

$$F_{\exp} = \left\{ f(t) = c \left( 1 - e^{-at} \right) \right\} \text{ and}$$

$$F_{\text{poly}} = \left\{ f(t) = c \left( 1 - b \left( t + b^{1/\rho} \right)^{-\rho} \right) \right\},$$

where $a, c, \rho \in (0, 1]$ and $b \in \mathbb{R} \geq 0$ are parameters whose values are selected randomly. These two families of functions are able to represent the general increasing-then-converging pattern of different shapes [33], where functions originating from $F_{\exp}$ exhibit

a rapid increase before converging, while those from $F_{\text{poly}}$ may display much slower growth rates. The generated reward functions of the 2-arm bandits and 15-arm bandits are shown in Figure 3a and 3c respectively. We introduce stochasticity by adding a Gaussian noise with a standard deviation of 0.1 in each reward observation.

*5.2.2 Results.* The results on 2-arm bandits and 15-arm bandits settings are shown in Figure 3b and 3d respectively, which plot the empirical cumulative regret over 200,000 iterations for 2-arm bandits and 1,500,000 iterations for 15-arm bandits.

*Answer to **RQ1**.* We can clearly observe that our proposed TI-UCB achieves the lowest regret at the horizon compared with all baselines. Though at the initial stage, some baselines such as SW-UCB outperform TI-UCB, they fail to explore sufficiently the optimal arm and converge to sub-optimal ones, resulting in linear regrets. From the results, we also observe that besides TI-UCB, the regret slope of SW-TS displays a trend of decreasing as well but the decrease is much later than TI-UCB. This indicates that SW-TS somehow reacts to the converging points of rewards but not as prompt as TI-UCB. Note that all algorithms except TI-UCB and R-ed-UCB do not have a theoretical guarantee on regret in such increasing-then-converging bandit setting. While R-ed-UCB has also guaranteed regret upper bound, it does not show consistent empirical advantages as TI-UCB does, which is also implied in [33]. We further compare their performances in real-world environments in Section 5.3 and 5.4.

## 5.3 Classification Model Selection

In this section, we evaluate the performance of TI-UCB compared with baselines on a classification model selection task.

*5.3.1 Data Generation.* We build a binary classification problem on IMDB dataset preprocessed as [32]. The IMDB dataset consists of 50,000 reviews and after preprocessing each review $x_t$ has a $d = 1,000$ dimensional feature. Each arm corresponds to an online optimization model for binary classification. Following [33], we formulate an IMDB 8-arm bandits by choosing:

- **LR(0.001)** and **LR(0.1)**: two online logistic regression models with learning rates of 0.001 and 0.1.
- **NB**: a naive bayes model.
- **AG(0.05)** and **AG(0.003)**: two adaptive gradient models with learning rates of 0.05 and 0.003.

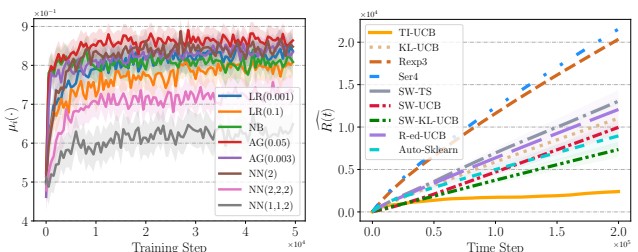

(a) IMDB Bandits: Reward Functions  (b) IMDB Bandits: Cumulative Regret

**Figure 4: Online selection of canonical classification models on IMDB datasets.**

- **NN(2)**, **NN(2,2,2)** and **NN(1,1,2)**: three neural network models, the first one consists of one layer with two nodes, the second one consists of three layers with two nodes in each layer, and the third one consists of three layers with one node in the first two layers and two nodes in the last layer.

In each round, a sample $x_t$ is firstly randomly selected from the dataset. Then the agent selects an arm, i.e. classification model, to predict make a prediction $\hat{y}_t$ on the sample and receives a binary reward $\mathbf{1}[\hat{y}_t = y_t]$ corresponding to whether the prediction is correct. After each arm pull, an online update is conducted to the chosen model with the selected sample $x_t$. Note that here is slight abuse of notation where we consider an online update as a finetuning step. We average 30 independent runs to visualize the reward trend of the above classification models as shown in Figure 4a. Note that while the general increasing-then-converging patterns are observed, they are subject to different extent of fluctuations, which poses further challenges for accurate and efficient online model selection.

*5.3.2 Results.* The results on classification model selection are shown in Figure 4b, which plots the empirical cumulative regrets over 200,000 iterations.

*Answer to **RQ1**.* As shown in Figure 4b, TI-UCB outperforms all considered baselines at the horizon with a considerable improvement even with fluctuations in the reward trends. In comparison, R-ed-UCB has not converged to the optimal arm yet at the horizon. While Auto-Sklearn is outperformed by TI-UCB, it shows some improvements over some other baselines such Ser4 and Rexp3, the extra computation cost and time due to iterative optimizations make it less efficient to be applied in online model selection. The results again demonstrate the empirical effectiveness of TI-UCB in online model selection with fluctuations in reward trends.

## 5.4 Large Language Model Selection

In this section, we present the experimental results on large language model selection for summarization task. Notably, we introduce the finetuning cost of API-based LLM into the reward design in addition to model performance to explore the economic tradeoffs of LLMs discussed in [21].

*5.4.1 Data Generate.* We choose text summarization as the downstream task with XSum [34] dataset. The XSum datasets contains

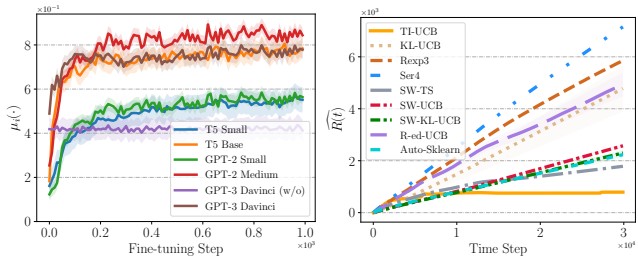

(a) LLM Bandits: Reward Functions  (b) LLM Bandits: Cumulative Regret

**Figure 5: Online selection of large language models on XSum datasets for summarization.**

204,045 samples of text-summary pairs. Each arm corresponds to an API-based LLM or a local small LLM selected as follows:

- **T5 Small** [40]: a small version of the text-to-text transfer transformer model with 60 million parameters.
- **T5 Base** [40]: a base-sized version of the text-to-text transfer transformer model with 220 million parameters.
- **GPT-2 Small** [39]: a small version of the GPT-2 model with 117 million parameters.
- **GPT-2 Medium** [39]: a medium-sized version of the GPT-2 model with 355 million parameters.
- **GPT-3 Davinci** [6]: an API-based large language model hosted by OpenAI with 175 billion parameters.

Similarly to Figure 1, in each round, a random batch of documents is selected from the dataset. Then, the agent chooses an LLM to summarize the documents. The quality of generated summaries are then evaluated using ROUGE-2 [29] score by comparing them with the reference summaries. After each arm pull, the chosen LLM is finetuned with the batch of samples. For local small LLMs, We set the batch size to be 16. AdamW optimizer is used for fine-tuning and the learning rate is set to be $5e^{-5}$. The fine-tuning processes run on four NVIDIA RTX2080Ti GPUs. For API-based LLMs, we use the API finetuning hosted by OpenAI. In addition, we include a zero-shot version of GPT-3 without the finetuning step as another base model to be compared, i.e. **GPT-3-davinci (w/o)**.

Instead of directly using ROUGE-2 as the reward, we introduce the finetuning cost of API-based model. Specifically, if the chosen is GPT-3 Davinci, the reward is constructed as

$$X_t = \text{ROUGE-2} - \eta_t ,$$

where $X_t$ is the reward and $\eta_t$ represents the monetary cost of finetuning, which is calculated as the cumulative sum $\eta_t = \eta_{t-1} + m \cdot \mathbf{1}[\text{Do Finetuning}]$ with $\eta_0 = 0$ where [Do Finetuning] is an indicator of whether we do finetuning at this step. For API-based LLM, we set $m = 0.01$ and for small local LLM, we set $m = 0.0001$. The values of finetuning cost $m$ selected are approximate values calculated based on API-finetuning rate per token charged by OpenAI and typical tokens length of 500 per document for text summarization task following [21]. To avoid excessive cumulation of finetuning costs, we stop the iterative finetuning after 100 consecutive finetuning steps without performance improvement over 0.1. The resulting

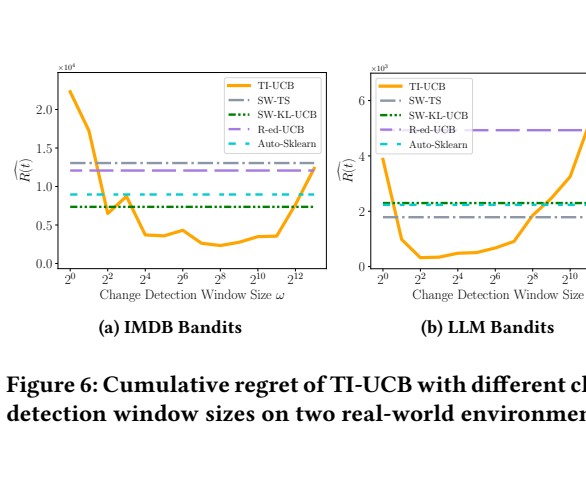

(a) IMDB Bandits      (b) LLM Bandits

**Figure 6: Cumulative regret of TI-UCB with different change detection window sizes on two real-world environments.**

normalized reward trends of considered LLMs are shown in Figure 5a, again showcasing the increasing-then-converging trends with some extent of fluctuations.

Note that we are fully aware that our reward design might not be the actual implementation as there is also cost of inference and different organizations may have different priorities in employing LLMs. Our primary focus in this paper is to propose a method that can promisingly handles the economic tradeoffs of LLM deployment, where more fine-grained and target-specific reward design related to LLMs can be easily plugged in.

*5.4.2 Results.* The results on large language model selection for text summarization on XSum dataset are shown in Figure 5b, which plots the empirical cumulative regrets over 30,000 iterations.

*Answer to* **RQ1.** From the results, we again observe the advantage of considering the increasing-then-converging trend with TI-UCB achieving the lowest cumulative regret. We also observe that the sliding-window-based algorithms, SW-TS, SW-UCB, and SW-KL-UCB, outperform most of other baselines, which shows the generalizability of sliding-window method in various non-stationary bandit problems.

*Answer to* **RQ2.** As illustrated in Figure 5a, the API-based GPT-3 Davinci achieves high performances at the initial stage even without finetuning while other small LLMs all perform poorly. However, after some finetuning steps, the model performances of T5 Base and GPT-2 Medium increases rapidly and are comparable with the performance of GPT-3 Davinci. Meanwhile, the reward of GPT-3 Davinci first increases and then quickly converges, which represents that the performance improvement brought by finetuning may not surpass the monetary cost induced by finetuning. Thus the economics tradeoffs make the further finetuning of API-based LLM a sub-optimal option compared to finetuning small LLMs. From Figure 5b, we can observe that our TI-UCB algorithm captures the increasing-then-converging rewards trend of GPT-3 Davinci due to the economic tradeoffs and predicts the potential reward increase of GPT-2 Medium. As a result, instead of being trapped in the initial high reward of API-based LLM, TI-UCB effectively and efficiently explores the best-performing LLM and makes the optimal model selection with a small amount of reward observations.

## 5.5 Different Change Detection Window Size

As shown in Section 5.3 and 5.4, TI-UCB has demonstrated its effectiveness when encountering increasing-then-converging reward trends with fluctuations in real-world environments. In this seciton, we further analyze the effectiveness of the change detection mechanism in the two real world environments constructed above. Specifically, we vary the change detection window size $\omega$ from $2^0$ to $2^{13}$ and evaluate the performances TI-UCB in both classification model selection and LLM selection environments. The results are shown in Figure 6, where only four competitive baselines are selected and shown based on the performances from previous experiments due to limited space.

*Answer to* **RQ3.** From Figure 6, we observe that TI-UCB outperforms compared baselines with the change detection window size $\omega$ in a certain range. Specifically, TI-UCB achieves the lowest regret with $\omega \in [2^4, 2^{11}]$ on IMDB bandits and with $\omega \in [2^1, 2^7]$ on LLM bandits. Though the performance of TI-UCB degenerates when $\omega$ is very small or very large, the wide ranges where TI-UCB performs best still demonstrate its robustness in terms of hyperparameter sensitivity and readiness for practical applications.

Analyzing the cases when $\omega$ is small, we attribute the performance degeneration to the fluctuations in the increasing-then-converging reward trend as observe in Figure 4a and 5a. As rewards typically do not change in a smooth or monotonic manner and may drastically go up and down in a short period, a small size of change detection window such as $2^0$ could easily result in false detection of change points. Moreover, comparing Figure 6a and Figure 6b, we can further observe a right skew of TI-UCB's regrets on IMDB bandits, which also conforms the observation from Figure 4a and 5a that IMDB bandits show more fluctuations and thus the window size needs to be larger to mitigate them. Such implication further suggests the selection of $\omega$ may be subject to the consideration of handling potential fluctuations in practice.

For large values of $\omega$ such as $2^{13}$, the reason of performance degeneration is due to the reduced frequency and latency of change detection. As $\omega$ approaches the evaluation horizon $T$, it is even possible that there are not sufficient amount of reward observations for change detection, which on the other hand, demonstrates the effectiveness and necessity of our change detection mechanism.

## 6 CONCLUSION

In this work, we have explored the pressing issue of online model selection, notably within the context of the rapidly evolving field of LLMs. By capitalizing on the increasing-then-converging pattern in model performance being trained or finetuned, we propose the TI-UCB algorithm, which can promising predict the reward increases and detect converging points with a sliding-window change detection mechanism. We theoretically prove an improvement of regret upper bound from prior work's polynomial regret to logarithmic in a similar setting. Extensive experiments also demonstrate empirically the advantage of TI-UCB in online model selection for canonical classification models and state-of-the-art LLMs. Our work underscores the necessity of considering the increasing-then-converging reward trend in online model selection, which paves the road for more efficient and economic model selection in the deployment of LLMs.

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

# APPENDIX

In Section A, we provide theoretical analysis of the concentration level of upper confidence bound and change detection criteria of our algorithm. In Section B, we present detailed proof of the regret upper bound of our algorithm. In Section C, we provide details of parameter setting of compared baselines.

## A  THEORETICAL ANALYSIS

### A.1  Proof for Concentration Inequalities

For each arm $i$, its reward satisfies a linear increase with the number of times it is pulled and then tends to a stationary state. Before we give the proof of concentration inequalities, we introduce a property of subgaussian variables following bandit algorithm theory.

**Lemma 1.** *If $X_i - \mu$ ($i = 1, 2, \cdots, n$) are independent $M$-subgaussian random variables, then for any $\varepsilon \geq 0$,*

$$\mathbb{P}\left(\hat{\mu} - \mu \geq \varepsilon\right) \leq \exp\left\{-\frac{n\varepsilon^2}{2M^2}\right\},$$

$$\mathbb{P}\left(\hat{\mu} - \mu \leq -\varepsilon\right) \leq \exp\left\{-\frac{n\varepsilon^2}{2M^2}\right\}, \tag{5}$$

*where $\hat{\mu} = \frac{\sum_{i=1}^{n} X_i}{n}$ is the expectation of independent variables [25].*

*Proof of Proposition 1.* Recall that $\mu_{i,1}, \mu_{i,2}, \cdots, \mu_{i,n}, (n_i(t) = n)$ are $n$ true reward values of arm $i$ during $[0, t]$, and $\hat{a}_i$ and $\hat{b}_i$ are the least squares estimations. For any $\varepsilon > 0$, the upper bound of the difference for the estimated values and true values are

$$\mathbb{P}\left(\hat{a}_i - a_i \geq \varepsilon\right) \leq \exp\{-\frac{n\varepsilon^2}{288}\}, \quad \mathbb{P}\left(\hat{b}_i - b_i \geq \varepsilon\right) \leq \exp\left\{-\frac{n\varepsilon^2}{128}\right\},$$

and

$$\mathbb{P}\left(\hat{\mu}_{i,n} - \mu_{i,n} \geq \varepsilon\right) \leq \exp\left\{-\frac{n\varepsilon^2}{128}\right\}.$$

Choose $\delta = \exp\{-\frac{n\varepsilon^2}{512}\} \in (0, 1)$, then

$$\mathbb{P}\left(\hat{\mu}_{i,n} - \mu_{i,n} \geq 16\sqrt{\frac{2\ln(1/\delta)}{n}}\right) \leq \delta. \tag{6}$$

Then for any $\delta \in (0, 1)$, $\mu \leq \hat{\mu} + 16\sqrt{\frac{2\ln(1/\delta)}{n}}$ holds with probability at least $1 - \delta$.

To demonstrate equation (6), we first explain the regression estimation of unknown terms of reward $\hat{a}_i, \hat{b}_i$. Denote $X_{i,s}$ as the observed reward of arm $i$ when arm $i$ has been pulled $s$ times ($s = 1, 2, \cdots, n$) with true value $\mu_{i,s}$. For $\hat{a}_i$, we have

$$\hat{a}_i - a_i = \sum_{s=1}^{n_i(t)} \frac{s - \bar{s}}{\sum_{s=1}^{n_i(t)} (s - \bar{s})^2} \left[X_{i,s} - \mu_{i,s}\right]$$

$$= \frac{1}{n} \sum_{s=1}^{n} \frac{12}{n^2 - 1} (s - \bar{s}) \left(X_{i,s} - \mu_{i,s}\right).$$

Due to the fact that the mean in the sum term is 0 and its absolute value has an upper bound of $\frac{12}{n^2-1}|s-\bar{s}||X_{i,s}-\mu_{i,s}| \leq \frac{12}{n^2-1} \cdot \frac{n+1}{2} \cdot 2 \leq 12$, then by Lemma 1, we obtain

$$\mathbb{P}\left(\hat{a}_i - a_i \geq \varepsilon\right) \leq \exp\left\{-\frac{n\varepsilon^2}{288}\right\}.$$

Similarly for $\hat{b}_i$,

$$\hat{b}_i - b_i = \sum_{s=1}^{n_i(t)} \left[\frac{1}{n_i(t)} - \frac{(s - \bar{s})\bar{s}}{\sum_{s=1}^{T_i(t)} (s - \bar{s})^2}\right] [X_{i,s} - \mu_{i,s}].$$

$$\mathbb{P}\left(\hat{b}_i - b_i \geq \varepsilon\right) = \mathbb{P}\left(\frac{1}{n}\sum_{s=1}^{n} Y_s \geq \varepsilon\right) \leq \exp\left\{-\frac{n\varepsilon^2}{128}\right\}.$$

Therefore, the estimated reward in the linear growth stage is

$$\hat{\mu}_i(t+1) = \hat{a}_i \cdot (n_i(t) + 1) + \hat{b}_i$$

$$= \frac{1}{n}\sum_{s=1}^{n}\left[1 + \frac{6}{n-1}(s - \frac{n+1}{2})\right] X_{i,s}.$$

$$\hat{\mu}_i(t+1) - \mu_i(t+1) \leq \frac{1}{n}\sum_{s=1}^{n}\left[1 + \frac{6}{n+1}(s - \frac{n+1}{2})\right]\left(X_{i,s} - \mu_{i,s}\right).$$

Similar result is obtained that

$$\mathbb{P}\left(\hat{\mu}_i(t+1) - \mu_i(t+1) \geq \varepsilon\right) \leq \exp\left(-\frac{n\varepsilon^2}{128}\right).$$

### A.2  Proof for Change Detection

Recall that each detection is performed by comparing the predicted reward at time step $t+1$ calculated based on the previous window of reward observations from $w_1 = [n_i(t) - 2\omega + 1, n_i(t) - \omega]$, and predicted reward at time $t+1$ inferred from the current window $w_2 = [n_i(t) - \omega + 1, n_i(t)]$, which we refer to as $\hat{\mu}_{w_1,A_t}(t+1)$ and $\hat{\mu}_{w_2,A_t}(t+1)$ respectively. As stated in Proposition 2, if the difference of the two predictions exceeds the preset threshold $\gamma/2$ as Line 10 in Algorithm 1, we consider a change point is detected and the reward observation records of arm $A_t$ will be re-initialized. Otherwise, the algorithm continues to pull arms, minimizing the regret while detecting change points with observations. In this section, we provide some analysis of the rationality of such criteria for detecting change points.

*A.2.1 Change does not happen.* For any given $\delta$ and for any $0 < \varepsilon_1 \leq \frac{\gamma}{2}$, if $\mathbb{P}\left(|\hat{\mu}_{w_1,i}(t+1) - \hat{\mu}_{w_2,i}(t+1)| \geq \varepsilon_1\right) \leq \delta$, then

$$\mathbb{P}\left(|\hat{\mu}_{w_1,i}(t+1) - \hat{\mu}_{w_2,i}(t+1)| \geq \frac{\gamma}{2}\right) \leq \delta$$

holds. In order to find proper $\varepsilon_1$, the least square method is used to represent the reward estimation at time step $t+1$.

$w_1$ *stage.* Denote $a_{w_1,i}$ and $b_{w_1,i}$ as the linear growth parameter and the interpret term of $w_1$ stage for arm $i$, respectively.

$$\hat{a}_{w_1,i} = \sum_{s=n_i(t)-2\omega+1}^{n_i(t)-\omega} \frac{s - \bar{s}}{\sum_{s=n_i(t)-2\omega+1}^{n_i(t)-\omega} (s - \bar{s})^2} X_s$$

$$\hat{b}_{w_1,i} = \sum_{s=n_i(t)-2\omega+1}^{n_i(t)-\omega} \left[\frac{1}{\omega} - \frac{(s - \bar{s})\bar{s}}{\sum_{s=n_i(t)-2\omega+1}^{n_i(t)-\omega} (s - \bar{s})^2}\right] X_s.$$

Since $\bar{s} = \frac{2n_i(t)-3\omega+1}{2}$, $\sum_{s=n_i(t)-2\omega+1}^{n_i(t)-\omega} (s - \bar{s})^2 = \frac{\omega(\omega^2-1)}{12}$, then we have

$$\hat{\mu}_{w_1,i}(t+1) = \hat{a}_{w_1,i} \cdot (n_i(t)+1) + \hat{b}_{w_1,i}$$

$$= \sum_{s=n_i(t)-2\omega+1}^{n_i(t)-\omega} \left[ \frac{6(s-\bar{s})(1+3\omega)}{\omega(\omega^2-1)} + \frac{1}{\omega} \right] X_s .$$

$w_2$ *stage.* Similarly, $a_{w_2,i}$ and $b_{w_2,i}$ means the linear growth term and the interpret term of $w_2$ stage, respectively.

$$\hat{a}_{w_2,i} = \sum_{s=n_i(t)-\omega+1}^{n_i(t)} \frac{s-\bar{s}}{\sum_{s=n_i(t)-2\omega+1}^{n_i(t)-\omega}(s-\bar{s})^2} X_s$$

$$\hat{b}_{w_2,i} = \sum_{s=n_i(t)-\omega+1}^{n_i(t)} \left[ \frac{1}{\omega} - \frac{(s-\bar{s})\bar{s}}{\sum_{s=n_i(t)-2\omega+1}^{n_i(t)-\omega}(s-\bar{s})^2} \right] X_s .$$

Similarly, we have

$$\hat{\mu}_{w_2,i}(t+1) = \hat{a}_{w_2,i} \cdot (n_i(t)+1) + \hat{b}_{w_2,i}$$

$$= \sum_{s=n_i(t)-\omega+1}^{n_i(t)} \left[ \frac{6(s-\bar{s})}{\omega(\omega+1)} + \frac{1}{\omega} \right] X_s .$$

Then for $\hat{\mu}_{w_1,i}(t+1) - \hat{\mu}_{w_2,i}(t+1)$,

$$\hat{\mu}_{w_1,i}(t+1) - \hat{\mu}_{w_2,i}(t+1)$$

$$= \frac{1}{\omega} \sum_{s=n_i(t)-\omega+1}^{n_i(t)} \left\{ \left[ \frac{6(s-\bar{s})(1+3\omega)}{\omega^2-1} + 1 \right] X_{s-\omega} \right.$$

$$\left. - \left[ \frac{6(s-\bar{s})}{\omega+1} + 1 \right] X_s \right\} .$$

Since $\hat{\mu}_{w_2,i}(t+1) - \mu_{w_2,i}(t+1)$ is a sub-gaussian variable as its mean is zero and $\mu_{w_1,i}(t+1) = \mu_{w_2,i}(t+1)$ when change has not happened, we have

$$\hat{\mu}_{w_1,i}(t+1) - \hat{\mu}_{w_2,i}(t+1) = \left[ \hat{\mu}_{w_1,i}(t+1) - \mu_{w_1,i}(t+1) \right]$$

$$+ \left[ \mu_{w_1,i}(t+1) - \mu_{w_2,i}(t+1) \right] + \left[ \mu_{w_2,i}(t+1) - \hat{\mu}_{w_2,i}(t+1) \right] .$$

Let $Z_s = \left[ \frac{6(s-\bar{s})(1+3\omega)}{\omega^2-1} + 1 \right] X_{s-\omega}$ and $Y_s = \left[ \frac{6(s-\bar{s})}{\omega+1} + 1 \right] X_s$. Similar to the proof in A.1, $|Z_s + Y_s|$ has an upper bound $\left( 14 + \frac{12}{|\omega-1|} \right)$. By Lemma 1, we obtain

$$\mathbb{P}\left( |\hat{\mu}_{w_1,i}(t+1) - \hat{\mu}_{w_2,i}(t+1)| \geq \varepsilon_1 \right) \leq 2 \cdot \exp\left( -\frac{\omega \varepsilon_1^2}{2(14 + \frac{12}{|\omega-1|})^2} \right) . \tag{7}$$

Choose $\delta_0 = 2 \cdot \exp\left( -\frac{\omega \varepsilon_1^2}{2\left(14 + \frac{12}{|\omega-1|}\right)^2} \right)$, and we can obtain $\varepsilon_1 = \sqrt{\frac{2}{\omega} \left( 14 + \frac{12}{|\omega-1|} \right)^2 \ln(\frac{2}{\delta_0})}$[1]. Therefore, for any given $\delta_0$, if $\gamma$ satisfies $\frac{\gamma}{2} \geq \varepsilon_1 = \sqrt{\frac{2}{\omega}(14 + \frac{12}{|\omega-1|})^2 \ln(\frac{2}{\delta_0})}$, then we have

$$\mathbb{P}\left( \left| \hat{\mu}_{w_1,i}(t+1) - \hat{\mu}_{w_2,i}(t+1) \right| \geq \frac{\gamma}{2} \right) \leq \delta_0 . \tag{8}$$

[1]Note the upper bound in equation 7 is not the minimum upper bound, thus the selection of $\varepsilon_1$ is not unique.

*A.2.2 Change happens.* When change happens, the similar idea is to find an $\varepsilon_2$, such that for any $\frac{\gamma}{2} \leq \varepsilon_2$,

$$\mathbb{P}\left( |\hat{\mu}_{w_1,i}(t+1) - \hat{\mu}_{w_2,i}(t+1)| \geq \frac{\gamma}{2} \right) \geq 1 - \delta_0 . \tag{9}$$

With $\varepsilon_1 \leq \frac{\gamma}{2} \leq \varepsilon_2$ holds, we can get a simple way to choose $\gamma$ as

$$\varepsilon_1 = \varepsilon_2 = \frac{\gamma}{2} . \tag{10}$$

Such threshold $\gamma/2$ is capable to identify whether the reward has reached a stable state.

Different from Section A.2.1, $\mu_{w_1,i}(t+1) = \mu_{w_2,i}(t+1)$ does not hold when change happens. Consider the assumption of linear growth of reward, which means that $|\mu_{w_1,i}(t+1) - \mu_{w_2,i}(t+1)|$ is approximately equal to $a_i \cdot \omega$. Thus, when change happens, we have

$$\left| \hat{\mu}_{w_1,i}(t+1) - \hat{\mu}_{w_2,i}(t+1) \right|$$

$$= |\hat{\mu}_{w_1,i}(t+1) - \mu_{w_1,i}(t+1) + \mu_{w_1,i}(t+1)$$

$$- \mu_{w_2,i}(t+1) + \mu_{w_2,i}(t+1) - \hat{\mu}_{w_2,i}(t+1)|$$

$$> |\mu_{w_1,i}(t+1) - \mu_{w_2,i}(t+1)| - |\hat{\mu}_{w_1,i}(t+1) - \mu_{w_1,i}(t+1)|$$

$$- |\hat{\mu}_{w_2,i}(t+1) - \mu_{w_2,i}(t+1)|$$

$$= a \cdot \omega - |\hat{\mu}_{w_1,i}(t+1) - \mu_{w_1,i}(t+1)| - |\hat{\mu}_{w_2,i}(t+1) - \mu_{w_2,i}(t+1)|$$

$$> \gamma - |\hat{\mu}_{w_1,i}(t+1) - \mu_{w_1,i}(t+1)| - |\hat{\mu}_{w_2,i}(t+1) - \mu_{w_2,i}(t+1)| .$$

When the change does not occur, then the difference between the predicted values of the two windows at time $t+1$ is smaller than $\gamma/2$, we claim that the difference between the estimated value and the true value will not be greater than $\gamma/4$ at this time with the following lemma.

**Lemma 2.** *For change detections,* $\left| \hat{\mu}_{w_1,i}(t+1) - \mu_{w_1,i}(t+1) \right| < \frac{\gamma}{4}$ *and* $\left| \hat{\mu}_{w_2,i}(t+1) - \mu_{w_2,i}(t+1) \right| < \frac{\gamma}{4}$ *hold when* $a \cdot \omega > \frac{\gamma}{2}$.

Proof. By Equation A.1, for any $\varepsilon_{pred}$ and a given $\delta > 0$, if

$$\mathbb{P}\left( \hat{\mu}_{w_1,i}(t+1) - \mu_{w_1,i}(t+1) \geq \varepsilon_{pred} \right) \leq \exp\left( -\frac{\omega \varepsilon_{pred}^2}{128} \right) = \delta ,$$

then $\varepsilon_{pred} = \sqrt{\frac{128}{\omega} \ln(1/\delta)}$. This implies that $\hat{\mu}_{w_1,i}(t+1) - \mu_{w_1,i}(t+1) < \sqrt{\frac{128}{\omega} \ln(1/\delta)}$ a.s. for sufficiently small $\delta$. The conclusion also holds for $w_2$. In addition, by equation 7, for any $\varepsilon_{CD} > 0$ and a given $\delta$

$$\mathbb{P}\left( \hat{\mu}_{w_1,i}(t+1) - \hat{\mu}_{w_2,i}(t+1) \geq \varepsilon_{CD} \right) \leq \exp\left( -\frac{\omega \varepsilon_{CD}^2}{2(14 + \frac{12}{|\omega-1|})^2} \right) = \delta,$$

then $\varepsilon_{CD} = \sqrt{\frac{2(14 + \frac{12}{|\omega-1|})^2}{\omega} \ln(1/\delta)}$, and $\hat{\mu}_{w_1,i}(t+1) - \hat{\mu}_{w_2,i}(t+1) < \sqrt{\frac{2\left(14 + \frac{12}{|\omega-1|}\right)^2}{\omega} \ln(1/\delta)}$ almost everywhere. We can always choose $\omega$ to make the $\varepsilon_{CD}$ at least twice as large as $\varepsilon_{LSE}$. Therefore, $\left| \hat{\mu}_{w_1,i}(t+1) - \mu_{w_1,i}(t+1) \right| < \frac{\gamma}{4}$, $\left| \hat{\mu}_{w_2,i}(t+1) - \mu_{w_2,i}(t+1) \right| < \frac{\gamma}{4}$ can always hold when $a \cdot \omega > \frac{\gamma}{2}$ and change has not happened. □

Thus with Lemma 2, when change has not happened, we have

$$|\hat{\mu}_{w_1,i}(t+1) - \hat{\mu}_{w_2,i}(t+1)|$$
$$= |\hat{\mu}_{w_1,i}(t+1) - \mu_{w_1,i}(t+1) + \mu_{w_1,i}(t+1) - \mu_{w_2,i}(t+1)$$
$$+ \mu_{w_2,i}(t+1) - \hat{\mu}_{w_2,i}(t+1)|$$
$$= |\hat{\mu}_{w_1,i}(t+1) - \mu_{w_1,i}(t+1) + \mu_{w_2,i}(t+1) - \hat{\mu}_{w_2,i}(t+1)|$$
$$\leq |\hat{\mu}_{w_1,i}(t+1) - \mu_{w_1,i}(t+1)| + |\hat{\mu}_{w_2,i}(t+1) - \mu_{w_2,i}(t+1)|$$
$$\leq \gamma/4 + \gamma/4 = \gamma/2.$$

Then

$$|\hat{\mu}_{w_1,i}(t+1) - \hat{\mu}_{w_2,i}(t+1)|_{\text{when change happens}}$$
$$\geq a \cdot \omega - |\hat{\mu}_{w_1,i}(t+1) - \mu_{w_1,i}(t+1)| - |\hat{\mu}_{w_2,i}(t+1) - \mu_{w_2,i}(t+1)|$$
$$> \gamma - \frac{\gamma}{4} - \frac{\gamma}{4} = \gamma/2 = \gamma/4 + \gamma/4$$
$$\geq |\hat{\mu}_{w_1,i}(t+1) - \hat{\mu}_{w_2,i}(t+1)|_{\text{when change has not happened}}$$

With the above analysis, we provide the rationality of the criteria for detecting change points:

(i) If $|\hat{\mu}_{w_1,i}(t+1) - \hat{\mu}_{w_2,i}(t+1)| > \frac{\gamma}{2}$, we think change happens.
(ii) If $|\hat{\mu}_{w_1,i}(t+1) - \hat{\mu}_{w_2,i}(t+1)| < \frac{\gamma}{2}$, we think change has not happened.

## B  PROOF FOR REGRET BOUND

In this section, we proof the regret upper bound of our algorithm present in Theorem 1. Recall that $F_i = \{\tau'_i > \nu_i\}$ and $D_i = \{\tau'_i \leq \nu_i + \omega\}$ indicate that the algorithm identifies changes before the actual change point occurs, and the algorithm does not detect changes within the window length after the actual change point, respectively, where $\nu_i$ denotes the moment that change happens, and $\tau'_i$ is the moment that our algorithm detects the change point successfully. When $F_1^c$ occurs, it means that the change point is detected by the algorithm, but in fact it does not happen, and the probability of $F_1^c$ will be bounded by $\mathbb{P}\left(F_1^c\right) = \mathbb{P}\left(\tau'_1 < \nu_1\right) = \mathbb{P}\left(|\hat{\mu}_1(t+1) - \hat{\mu}_2(t+1)| > \frac{\gamma}{2}\right) \leq \delta_0$ as mentioned before. We divide the $[0,T]$ stage into $[0,\nu_1]$ and $[\nu_1,T]$. In the $[0,\nu_1]$ stage, there is no arm reaching the threshold and this stage is a generalization of the standard UCB process except the distribution of the reward. The expected regret can be decomposed by:

$$\mathbb{E}[R(T)] = \mathbb{E}[R(T)\mathbb{1}\{F_1\}] + \mathbb{E}[R(T)\mathbb{1}\{F_1^c\}]$$
$$\leq \mathbb{E}[R(\nu_1)\mathbb{1}\{F_1\}] + \mathbb{E}[R(T) - R(\nu_1)] + T \cdot \mathbb{P}\left(F_1^c\right).$$

To prove the regret bound provided in Theorem 1, we prove the following lemmas in Section 4.2.

**Lemma 3.** *Before $\nu_1$, the number of times that arm $i$ is pulled is at most $2048\ln(1/\delta)/\Delta_i^2(t)$ in TI-UCB algorithm, where $\Delta_i(t)$ is the difference between the optimal reward and reward obtained by arm $i$ at time $t$.*

PROOF. For simplicity, we assume that $\bar{\mu}_i(t)$ is the upper confidence value (Equation 1 of $\text{UCB}_i(t-1,\delta)$ in Proposition 1) of arm $i$ at time $t$ and $\bar{\mu}_{i^*}(t)$ is the value of the optimal arm at time $t$. By

concentration inequality, we have

$$\mu_i(t) - 16\sqrt{\frac{2\ln(1/\delta)}{n_i(t)}} < \hat{\mu}_i < \mu_i(t) + 16\sqrt{\frac{2\ln(1/\delta)}{n_i(t)}}, \qquad (a)$$

By our definition of $\text{UCB}_i(t-1,\delta)$, we have

$$\bar{\mu}_i(t) = \hat{\mu}_i(t) + 16\sqrt{\frac{2\ln(1/\delta)}{n_i(t)}}. \qquad (b)$$

If arm $i$ is selected rather than arm $i^*$ at time $t$, it implies that

$$\bar{\mu}_i(t) > \bar{\mu}_{i^*}(t). \qquad (c)$$

Apply (a) to arm $i$ and arm $i^*$, then we have

$$\begin{cases} \hat{\mu}_i < \mu_i(t) + 16\sqrt{\dfrac{2\ln(1/\delta)}{n_i(t)}} \\[4mm] \hat{\mu}_{i^*} > \mu_{i^*}(t) - 16\sqrt{\dfrac{2\ln(1/\delta)}{n_i(t)}}. \end{cases} \qquad (d)$$

Hence, we can obtain

$$\mu_i(t) + 2\left[16 \cdot \sqrt{\frac{2\ln(1/\delta)}{n_i(t)}}\right]$$
$$\overset{(a)}{>} \hat{\mu}_i(t) + 16 \cdot \sqrt{\frac{2\ln(1/\delta)}{n_i(t)}}$$
$$\overset{(c)}{>} \hat{\mu}_{i^*}(t) + 16 \cdot \sqrt{\frac{2\ln(1/\delta)}{n_{i^*}(t)}}$$
$$\overset{(d)}{>} \hat{\mu}_{i^*}(t) - 16 \cdot \sqrt{\frac{2\ln(1/\delta)}{n_{i^*}(t)}} + 16 \cdot \sqrt{\frac{2\ln(1/\delta)}{n_{i^*}(t)}}$$
$$= \mu_{i^*}(t),$$

which leads to $n_i(t) < \frac{2048\ln(1/\delta)}{\Delta_i^2(t)}$, where $\Delta_i(t)$ means the difference between the optimal reward and reward obtained by arm $i$ at time $t$. Then for $\Delta_{min} = \min_{t\in[0,T],i\neq i^*}\{\mu_{i^*}(t) - \mu_i(t)\}$,

$$n_i(T) < \frac{2048\ln(1/\delta)}{\Delta_{min}^2}.$$

The $\Delta_{min}$ defined here does not change with the order in which the arms are pulled, it depends on the optimal policy hence it is unique. Therefore, the number of arm $i$ has been pulled before the moment that the first threshold is reached is at most $\frac{2048\ln(1/\delta)}{\Delta_{min}^2}$.  □

**Lemma 4.** *For $K > 0$, $T > 0$ and $\delta = O(1/T)$, we run the TI-UCB in $[0,T]$ before the moment that the first threshold is reached. Then the regret is at most*

$$\sum_{i:n_i(T)>n_i^*(T)} c_i \frac{2048\ln(T)}{\Delta_{min}^2} + K\left(\frac{\pi^2}{3}+1\right), \qquad (11)$$

*where $\Delta_{min} = \min_{t\in[0,T],i\neq i_t^*}\{\mu_{i_t^*}(t) - \mu_i(t)\}$ is the minimum gap between the optimal reward and the true reward.*

PROOF. Notice that there is no regret when $n_i(T) < n_i^*(T)$. The expected regret can be bounded by

$$\mathbb{E}\left[R(T)\right] \leq \mathbb{E}\left[\sum_{i:n_i(T) \geq n_{i^*}(T)} \left| \sum_{s=1}^{n_{i^*}(T)} \mu_{i,s} - \sum_{s=1}^{n_i(T)} \mu_{i,s} \right| \right].$$

And we have

$$\mathbb{E}\left[R_i(T)\right]$$

$$\leq \mathbb{E}\left[ \mathbb{1}\{i : n_i(T) \geq n_{i^*}(T)\} \cdot \left| \sum_{s=1}^{n_{i^*}(T)} \mu_{i,s} - \sum_{s=1}^{n_i(T)} \mu_{i,s} \right| \right]$$

$$= \mathbb{E}\left( \mathbb{1}\{i : n_i(T) \geq n_{i^*}(T)\} \cdot \left| \sum_{s=1}^{n_{i^*}(T)} (a_i \cdot s + b_i) - \sum_{s=1}^{n_i(T)} (a_i \cdot s + b_i) \right| \right)$$

$$= \mathbb{E}\left[ \left| \sum_{s=n_{i^*}(T)}^{n_i(T)} (a_i \cdot s + b_i) \right| \right] \overset{a_i \cdot s + b_i \leq c_i}{\leq} \mathbb{E}\left[ c_i \cdot (n_i(T) - n_{i^*}(T)) \right]$$

$$= c_i \left[ \mathbb{E}\left[ (n_i(T) - n_{i^*}(T)) \right] \right].$$

By $n_i(T) \leq \frac{2048 \ln(1/\delta)}{\Delta_{min}^2}$, for any positive integer $M$, we have

$$n_i(T)$$

$$\leq M + \sum_{t=M+1}^{T} \mathbb{1}\{A_t = i, n_i(t-1) \geq M\}$$

$$\leq M + \sum_{t=M+1}^{T} \mathbb{1}\left\{ UCB_{i^*}(t-1,\delta) \leq UCB_i(t-1,\delta), n_i(t-1) \geq M \right\}$$

$$\leq M + \sum_{t=M+1}^{T} \mathbb{1}\left\{ \min_{0<s<t}\left( \hat{\mu}_{i^*}(s) + 16 \cdot \sqrt{\frac{2\ln(1/\delta)}{n_i^*(s)}} \right) \right.$$

$$\left. \leq \max_{M \leq h < t}\left( \hat{\mu}_i(h) + 16 \cdot \sqrt{\frac{2\ln(1/\delta)}{n_i(h)}} \right) \right\}$$

$$\leq M + \sum_{t=M+1}^{T} \sum_{s=1}^{t-1} \sum_{h=M}^{t-1} \left[ \hat{\mu}_{i^*}(s) + 16 \cdot \sqrt{\frac{2\ln(1/\delta)}{n_i^*(s)}} \right.$$

$$\left. \leq \hat{\mu}_i(h) + 16 \cdot \sqrt{\frac{2\ln(1/\delta)}{n_i(h)}} \right].$$

Note that $\hat{\mu}_{i^*}(s) + 16 \cdot \sqrt{\frac{2\ln(1/\delta)}{n_i^*(s)}} \leq \hat{\mu}_i(h) + 16 \cdot \sqrt{\frac{2\ln(1/\delta)}{n_i(h)}}$ holds when at least one of the following three inequalities is satisfied.

$$\begin{cases} \mu_{i^*}(s) \geq \hat{\mu}_{i^*}(s) + 16\sqrt{\frac{2\ln(1/\delta)}{n_i(s)}} & (e) \\\\ \mu_i(h) \leq \hat{\mu}_i(h) - 16\sqrt{\frac{2\ln(1/\delta)}{n_i(h)}} & (f) \\\\ \mu_{i^*}(s) < \mu_i(h) + 2*16\sqrt{\frac{2\ln(1/\delta)}{n_i(h)}} & (g) \end{cases}$$

Since inequality (g) does not occur for any $s \leq t-1$ when $n_i(t-1) \geq M$, we have

$$n_i(T) \leq M + \sum_{t=M+1}^{T} \sum_{s=1}^{t-1} \sum_{h=M}^{t-1} \left[ \mathbb{P}\left( \mu_{i^*}(s) \geq \hat{\mu}_{i^*}(s) + 16\sqrt{\frac{2\ln(1/\delta)}{n_i(s)}} \right) \right.$$

$$\left. + \mathbb{P}\left( \mu_i(h) \leq \hat{\mu}_i(h) - 16\sqrt{\frac{2\ln(1/\delta)}{n_i(h)}} \right) \right].$$

By concentration inequalities, the probability of (e) and the probability of (f) are less than $e^{-4\ln(T)} = T^{-4}$. Hence, $n_i(T) \leq M + \sum_{t=1}^{T} \sum_{s=1}^{t} \sum_{h=1}^{t} 2t^{-4}$. Choose $M = \lceil \frac{2048\ln(T)}{\Delta_{min}^2} \rceil$, then

$$n_i(T) \leq \frac{2048\ln(T)}{\Delta_{min}^2} + 1 + \frac{\pi^2}{3}.$$

Finally, the regret bound for UCB with the reward of each arm increasing linearly with time is

$$\mathbb{E}\left[R(\nu_1)\mathbb{1}\{F_1\}\right] = \mathrm{E}_{UCB}[\nu_1] \leq \sum_{i:n_i(\nu_1) \geq n_{i^*}(\nu_1)} c_i n_i(\nu_1)$$

$$\leq \sum_{i:n_i(T) \geq n_{i^*}(T)} c_i \frac{2048\ln(T)}{\Delta_{min}^2} + K\left(\frac{\pi^2}{3} + 1\right). \quad (12)$$

□

**Lemma 5** (Regret bound for $F_1^c$). *The probability of $P(F_1^c)$ can be upper bounded by a constant related to $T$.*

$$\mathbb{P}\left(F_1^c\right) = \mathbb{P}\left(\tau_1' < \nu_1\right) \leq \frac{2}{T}. \quad (13)$$

PROOF. Since $\mu_{w_1,i} = \mu_{w_2,i}$ when change has not happened,

$$\mathbb{P}\left(F_1^c\right) = \mathbb{P}\left(\tau_1' < \nu_1\right)$$

$$= \mathbb{P}\left( \left| \hat{\mu}_{w_1,i} - \hat{\mu}_{w_2,i} \right| > \frac{\gamma}{2} \right)_{\text{when change has not happened}}$$

$$= \mathbb{P}\left( \left| \hat{\mu}_{w_1,i} - \mu_{w_1,i} + \mu_{w_1,i} - \mu_{w_2,i} + \mu_2 - \hat{\mu}_{w_2,i} \right| > \frac{\gamma}{2} \right)$$

$$= \mathbb{P}\left( \left| (\hat{\mu}_{w_1,i} - \mu_{w_1,i}) + (\mu_{w_2,i} - \hat{\mu}_{w_2,i}) \right| > \frac{\gamma}{2} \right)$$

$$\underset{(h)}{\leq} \mathbb{P}\left( \left| \hat{\mu}_{w_1,i} - \mu_{w_1,i} \right| + \left| \mu_{w_2,i} - \hat{\mu}_{w_2,i} \right| > \frac{\gamma}{2} \right)$$

$$\underset{(i)}{\leq} \mathbb{P}\left( \left| \hat{\mu}_{w_1,i} - \mu_{w_1,i} \right| > \frac{\gamma}{4} \right) + \mathbb{P}\left( \left| \mu_{w_2,i} - \hat{\mu}_{w_2,i} \right| > \frac{\gamma}{4} \right)$$

$$\leq 2\delta^4 < 2\delta.$$

The inequality (h) can be considered as the following. Assume that the event $\mathcal{A}_1$ stands for $\left| (\hat{\mu}_{w_1,i} - \mu_{w_1,i}) + (\mu_{w_2,i} - \hat{\mu}_{w_2,i}) \right| > \frac{\gamma}{2}$, and $\mathcal{A}_2$ stands for $\left| \hat{\mu}_{w_1,i} - \mu_{w_1,i} \right| + \left| \mu_{w_2,i} - \hat{\mu}_{w_2,i} \right| > \frac{\gamma}{2}$. When $\mathcal{A}_1$ holds, $\mathcal{A}_2$ also holds. However, if $\mathcal{A}_2$ holds, $\mathcal{A}_1$ may not hold. An example of such case would be when $\hat{\mu}_{w_1,i} - \mu_{w_1,i} = \frac{\gamma}{2}$ and $\hat{\mu}_{w_2,i} - \mu_{w_2,i} = -\frac{\gamma}{4}$.

The inequality (i) can be proved similarly. Let $\mathcal{A}_3$ represents the event when at least one of the two inequalities $\left| \hat{\mu}_{w_1,i} - \mu_{w_1,i} \right| > \frac{\gamma}{4}$ and $\left| \mu_{w_2,i} - \hat{\mu}_{w_2,i} \right| > \frac{\gamma}{4}$ holds and we have

$$\mathbb{P}\left(\mathcal{A}_3\right) = \mathbb{P}\left( \left\{ \left| \hat{\mu}_{w_1,i} - \mu_{w_1,i} \right| > \frac{\gamma}{4} \right\} \cup \left\{ \left| \mu_{w_2,i} - \hat{\mu}_{w_2,i} \right| > \frac{\gamma}{4} \right\} \right)$$

$$= \mathbb{P}\left( \left| \hat{\mu}_{w_1,i} - \mu_{w_1,i} \right| > \frac{\gamma}{4} \right) + \mathbb{P}\left( \left| \mu_{w_2,i} - \hat{\mu}_{w_2,i} \right| > \frac{\gamma}{4} \right).$$

And $\mathcal{A}_2 \subset \mathcal{A}_3$ can be easily seen.

With Lemma 5 and the assumption that $\delta \leq \frac{1}{T}$, we can obtain
$$T \cdot \mathbb{P}\left(F_1^c\right) \leq 2. \qquad \square$$

**Lemma 6** (Regret bound from $\nu_1$ to $T$).

$$\mathbb{E}\left[R(T) - R(\nu_1)\right]$$
$$\leq \mathbb{E}\left[R(T) - R(\nu_1)|F_1D_1\right] + T \cdot (1 - P(F_1D_1))$$
$$= \mathbb{E}\left[R(T) - R(\tau_1')|F_1D_1\right] + \mathbb{E}\left[R(\tau_1') - R(\nu_1)|F_1D_1\right]$$
$$+ T \cdot (1 - P(F_1D_1)) \qquad (14)$$
$$\leq \cdots \leq \mathbb{E}\left[R(T) - R(\tau_K')\right] + K\omega + 2KL$$
$$\leq \sum_{i:n_i(T)>n_i^*(T)} c_i \frac{2048 \ln(T)}{\Delta_{\min}^2} + K\left(\frac{\pi^2}{3} + 1\right) + K\omega + 2KL,$$

where $L$ is a constant strictly less than $\ln T$.

PROOF. The regret of period $[\nu_1, T]$ can be decomposed by

$$\mathbb{E}\left[R(T) - R(\nu_1)\right]$$
$$\leq \mathbb{E}\left[R(T) - R(\nu_1)|F_1D_1\right] + T \cdot (1 - P(F_1D_1))$$
$$\leq \underbrace{\mathbb{E}\left[R(T) - R(\tau_1')|F_1D_1\right]}_{(i)} + \underbrace{\mathbb{E}\left[R(\tau_1') - R(\nu_1)|F_1D_1\right]}_{(ii)}$$
$$+ \underbrace{T(1 - \mathbb{P}(F_1D_1))}_{(iii)} .$$

Next, we analyze these three parts separately.
**Part(iii)**:
$$\mathbb{P}(F_1D_1) = \mathbb{P}\left(\nu_1 < \tau_1' < \nu_1 + \omega\right) .$$

This means that if a change occurs, the algorithm can successfully detect it within the window width $\omega$. Then by inequality 9 we have

$$\mathbb{P}(F_1D_1) = \mathbb{P}\left(\left|\hat{\mu}_{w_1,i}(t+1) - \hat{\mu}_{w_2,i}(t+1)\right| > \frac{\gamma}{2}\right) > 1 - \delta_0 .$$

Note that $\delta_0 \leq L\delta$ for a constant $L < \ln(\frac{1}{\delta})$ and $\delta \leq \frac{1}{T}$. When change has not happened, we have

$$\mathbb{P}\left(\left|\hat{\mu}_i - \mu_i\right| \geq \frac{\gamma}{2}\right) \leq \exp\left\{-\frac{\omega(\frac{\gamma}{2})^2}{128}\right\} = \delta,$$
$$\mathbb{P}\left(\left|\hat{\mu}_{w_1,i} - \hat{\mu}_{w_2,i}\right| \geq \frac{\gamma}{2}\right) \leq \exp\left\{-\frac{\omega(\frac{\gamma}{2})^2}{2(14 + \frac{12}{|\omega-1|})^2}\right\} = \delta_0 .$$

Then we obtain $\delta_0 = \delta^{\frac{28}{7+\frac{6}{|\omega-1|^2}}}$. When $|\omega - 1| \geq \sqrt{\frac{\ln(L\delta)}{\frac{14}{3}\ln \delta - \frac{7}{6}\ln(L\delta)}}$, $\delta_0 \leq L\delta$ holds, and when $T$ is sufficiently large, $\frac{1}{\delta} \geq T$ is large, the condition for $|\omega - 1| \geq \sqrt{\frac{\ln(L\delta)}{\frac{14}{3}\ln \delta - \frac{7}{6}\ln(L\delta)}}$ holds naturally. Therefore, we have

$$1 - \mathbb{P}(F_1D_1) < \delta_0 \leq L\delta \leq \frac{2L}{T} .$$

Thus Part(iii) $\leq 2L < 2 \ln(T)$ holds.
**Part(ii)**:

We have
$$\mathbb{E}\left[R(\tau_1') - R(\nu_1)|F_1D_1\right] \leq \mathbb{E}\left[\tau_1' - \nu_1|F_1D_1\right]$$
$$\leq \mathbb{E}\left[\tau_1' - \nu_1|\nu_1 < \tau_1' \leq \nu_1 + \omega\right]$$
$$\leq \omega .$$

**Part(i)**:
Let $\widetilde{E} = \mathbb{E}\left[\cdot|F_1D_1\right]$, then $\mathbb{E}\left[R(T) - R(\tau_1')|F_1D_1\right] = \widetilde{E}(R(T) - R(\tau_1)) \leq \widetilde{E}(R(T - \tau_1'))$.

Using recursive method, we have

$$\widetilde{E}(R(T - \tau_1'))$$
$$\leq \widetilde{E}(R(T - \tau_1')|F_2D_2) + T[1 - P(F_2D_2)]$$
$$= \widetilde{E}(R(T - \tau_2')|F_2D_2) + \widetilde{E}(R(\tau_2' - \nu_2)|F_2D_2) + T[1 - P(F_2D_2)]$$
$$= \widetilde{\widetilde{E}}(R(T - \tau_2')) + \widetilde{E}(R(\tau_2' - \nu_2)|F_2D_2) + T \cdot [1 - P(F_2D_2)]$$
$$\leq \widetilde{\widetilde{E}}(R(T - \tau_2')) + \omega + 2L$$
$$\leq \cdots$$
$$\leq \mathbb{E}\left[R(T - \tau_K')\right] + K\omega + 2KL .$$
$$\square$$

The stage of $[\tau_K', T]$ is a process similar to standard UCB with stationary reward, and its regret can also be bounded by

$$\mathbb{E}\left[R(T - \tau_K')\right] < \sum_{i:n_i(T)>n_i^*(T)} c_i \frac{2048 \ln(T)}{\Delta_{\min}^2} + K\left(\frac{\pi^2}{3} + 1\right) .$$

Then
$$\mathbb{E}\left[R(T - \tau_1')|F_1D_1\right]$$
$$\leq \sum_{i:n_i(T)>n_i^*(T)} c_i \frac{2048 \ln(T)}{\Delta_{\min}^2} + K\left(\frac{\pi^2}{3} + 1\right) + K\omega + 2KL . \qquad (15)$$

In summary, by Lemma 4, 5, and Equation 15, we obtain the final regret upper bound of our algorithm present in Theorem 1:

$$\mathbb{E}\left[R(T)\right]$$
$$= \mathbb{E}\left[R(T)\mathbb{1}\{F_1\}\right] + \mathbb{E}\left[R(T)\mathbb{1}\{F_1^c\}\right]$$
$$\leq \mathbb{E}\left[R(\nu_1)\mathbb{1}\{F_1\}\right] + T \cdot \mathbb{P}\left(F_1^c\right) + \mathbb{E}\left[R(T) - R(\nu_1)\right]$$
$$\leq \underbrace{\sum_{i:n_i(T)>n_i^*(T)} c_i \frac{2048 \ln(T)}{\Delta_{min}^2} + K\left(\frac{\pi^2}{3} + 1\right)}_{\text{Equation 12 in Lemma 4}} + \underbrace{2}_{\text{Lemma5}}$$
$$+ \underbrace{\sum_{i:n_i(T)>n_i^*(T)} c_i \frac{2048 \ln(T)}{\Delta_{\min}^2} + K\left(\frac{\pi^2}{3} + 1\right) + K\omega + 2KL}_{\text{Equation 15}}$$
$$= \sum_{i:n_i(T)>n_i^*(T)} c_i \frac{4096 \ln(T)}{\Delta_{\min}^2} + K\left(\frac{\pi^2}{3} + 1\right) + K\omega + 2KL + 2 .$$

## C PARAMETER SETTING OF BASELINE ALGORITHMS

We choose the parameters of compared baselines algorithms as follows:

- **KL-UCB**: $c = 3$ according to [17].
- **Rexp3**: $V_T = K$, $\gamma = \min\left\{1, \sqrt{\frac{K \log K}{(e-1)\Delta_T}}\right\}$, and
  $\Delta_T = \left\lceil (K \log K)^{1/3} (T/V_T)^{2/3} \right\rceil$ according to [3].
- **Ser4**: $\delta = 1/T$, $\epsilon = \frac{1}{KT}$, and $\phi = \sqrt{\frac{N}{TK} \log(KT)}$ according to [1].

- **SW-TS**: $\beta = 1/2$ and sliding-window $\tau = T^{1-\beta} = \sqrt{T}$ according to [46].
- **SW-UCB**: sliding-window $\tau = 4\sqrt{T \log T}$ and constant $\xi = 0.6$ according to [18].
- **SW-KL-UCB**: $\tau = \sigma^{-4/5}$ according to [10].
- **R-ed-UCB**: window parameter $\epsilon = 1/4$ for synthetic experiments and $\epsilon = 1/32$ for real-world experiments according to [33].

