# OpenReview forum: "Convergence-Aware Online Model Selection with Time-Increasing Bandits"
_ACM.org/TheWebConf/2024/Conference — TheWebConf24 Oral_

### Official Review · Reviewer_K1rg · 2023-11-20

**Novelty:** 5
**Technical Quality:** 5

**Review:**

The authors of the paper “Convergence-Aware Online Model Selection with Time-Increasing Bandits” present a novel framework for online model selection based on the multi-armed bandits formulation. The framework takes into account the property of loss convergence (or performance metric convergence) of learned models as part of the reward formulation. The authors support their proposed algorithm with both extensive experimentation and formal proofs.

In my opinion the work is very interesting and important given the present advancements in the LLMs field. The ability to effectively select a model from a pool of models in an online manner is highly desirable. The writing of the paper is clear and the overall presentation of results looks sound and the claims/RQs are well supported.

All in all, I think the experimental setup is well explained and easy to follow which increases the chances of reproducibility.

**Questions:**

Are you going to release the code upon acceptance? I think the experiments are well explained, however, the settings might be a bit complex in some cases.

**Ethics Review Description:**

.

**Reviewer Confidence:**

3: The reviewer is confident but not certain that the evaluation is correct

**Scope:**

4: The work is relevant to the Web and to the track, and is of broad interest to the community

---

### Official Review · Reviewer_vDTo · 2023-11-24

**Novelty:** 5
**Technical Quality:** 5

**Review:**

This paper studies the model selection problem tailored to the LLMs setting. In particular, it casts the problem as a variant of rest bandit problem with increasing reward. The increasing reward capture that the performance of an LLM would increase by fine tuning. TI-UCB algorithm is developed for this problem. Regret bounds are proved and extensive experiments are conducted to evaluate the TI-UCB. Overall, this paper is well written and has a fluent logic flow. It studies a timely and important problem. The algorithm analysis looks sound and experiments looks sufficient.

I appreciate this work, but I have several concerns on this paper.

The finetuning cost is frequently mentioned in the motivation and I am convinced that the cost is an important factor. However, it seems that the model does not capture the finetuning cost explicitly. The proposed model only captures the reward. Without capturing the finetuning cost, the proposed model is not well tailored to the problem.

The reward model that captures the convergence of LLM needs more justification. What’s the formal definition of the convergence of an LLM? Does an LLM model converge after several rounds of finetune? Under what conditions it converge? Intuitively, the convergence should depend on the finetune data. This rise a problem of how to select finetune data.

The regret definition is unclear. In particular, the n*_i(T) is not well defined. What do you mean by optimal action sequence in the definition of n*_i(T)? Intuitively, in different round, one may have different test document for finetuning, and the n*_i(T) should depends on the finetuning data. But it seems that it does not depend on the finetuning data.

**Questions:**

Please refer to my concerns.

**Ethics Review Description:**

NO.

**Reviewer Confidence:**

3: The reviewer is confident but not certain that the evaluation is correct

**Scope:**

4: The work is relevant to the Web and to the track, and is of broad interest to the community

---

### Official Review · Reviewer_bFMa · 2023-12-01

**Novelty:** 6
**Technical Quality:** 6

**Review:**

This paper focuses on the online model selection problem, where the models are fine-tuned continuously as when they receive feedback and show increasing-then-converging reward behavior. In this setting, the authors propose a time-increasing bandit algorithm, TI- UCB, which essentially predicts the increase of candidate model performances via finetuning and handles the usual exploration-exploitation scenarios in the bandit setting (for the model selection problem). The main novelty of the proposed algorithm comes from the change detection mechanism, where the authors compare consecutive increase predictions. The authors provide theoretical bounds on the cumulative regret which improves in comparison to the state-of-the-art baseline. Experiments on both simulated and real-world setting are provided which show the efficacy of the proposed algorithm. The application on selection of LLMs is also interesting.

Pros:
1. Relevant paper for the community
2. Contains appropriate theoretical and experimental contributions.
3. Solid presentation and does not require much effort for understanding.

Cons: I enjoyed reading the paper and do not have cons. Only a minor one stated below:
1. (Minor) Presentation can be improved by removing imprecise statements.

**Questions:**

Overall, I found the paper to be very useful for the community. It correctly highlights its importance via theoretical and experimental evaluation. The presentation of the paper is also solid. I have a few questions on which the authors can rebut on.

1. I do not know if the increasing-then-converging is a terminology used anywhere else. I believe the authors want to state the sublinear curve, right? Or, do the authors still want to use increasing-then-converging terminology?
2. In figure 2, is the reward shown the cumulative reward of the banding setting (eq 2) or the instantaneous reward of the models? I believe it’s the latter.
3. Line 358-359, the authors mention that v_i is the convergence point when the reward become “stable”. The word stable is imprecise. The authors should mention what do they mean by stable more formally.
4. Theorem 1 mentions that the regret becomes logarithmic in T; whereas, R-ed-UCB’s regret is polynomial in T. Does this really make the algorithm efficient since Theorem 1 also mentions that the algorithm’s regret is polynomial in (1/\Delta_min), and (1/\Delta_min) again depends on T due to non-stationarity. Can the authors comment on that?
5. How did the authors choose the m values for smaller models (in line 805)? Is it really 100 times cheaper to fine tune smaller models than the API based models?
6. In Figure 5(b), I see a sharp convergence of regret for TI-UCB. Can the authors comment on the time-step difference between the convergence of models’ rewards vs convergence of regret by T1-UCB? Or, are they not correlated?


Minor:
1. In eq1, it should be x_{A_s, s} instead of the random variable X itself.
2. Line 390, “and thus the cumulative regret” -> “and thus to minimize the cumulative regret”

**Reviewer Confidence:**

4: The reviewer is certain that the evaluation is correct and very familiar with the relevant literature

**Scope:**

4: The work is relevant to the Web and to the track, and is of broad interest to the community

---

### Official Review · Reviewer_87is · 2023-12-01

**Novelty:** 4
**Technical Quality:** 4

**Review:**

Pros:
1.  It is novel to predict the reward increases and detect converging points with a sliding-window change detection mechanism.
2. The authors theoretically prove the lower regret upper bound in their method.
Cons:
1. More backgrounds are needed for online model selection.
2. The experiments are conducted in a synthetic environment. Real online experiments are needed to make the results convincing.
3. It is not representative enough to verify the online model selection strategy on the text summarization task.

**Questions:**

Do the candidate models influence the model selection?

**Reviewer Confidence:**

3: The reviewer is confident but not certain that the evaluation is correct

**Scope:**

3: The work is somewhat relevant to the Web and to the track, and is of narrow interest to a sub-community

---

### Decision · Program_Chairs · 2024-01-22

**Decision:**

Accept (Oral)

**Comment:**

Authors present a work for online model selection using bandits augmented with LLMs. The work is novel, and highly relevant to the community. The paper is also technically sound, and the authors demonstrated the practical value on real-world datasets. Authors have satisfactory responses to reviewer questions, and all reviewers have acknowledged the author rebuttals.